# Relation between Alpha-Synuclein and Core CSF Biomarkers of Alzheimer’s Disease

**DOI:** 10.3390/medicina57090954

**Published:** 2021-09-10

**Authors:** Victoria Monge-García, María-Salud García-Ayllón, Javier Sáez-Valero, José Sánchez-Payá, Francisco Navarrete-Rueda, Jorge Manzanares-Robles, Ruth Gasparini-Berenguer, Raquel Romero-Lorenzo, María Angeles Cortés-Gómez, José-Antonio Monge-Argilés

**Affiliations:** 1Rehabilitation Service, University General Hospital of Alicante, 03010 Alicante, Spain; victoria_mgar@hotmail.com; 2Research Unit, University General Hospital of Elche, FISABIO, 03456 Elche, Spain; j.saez@umh.es; 3Instituto de Neurociencias de Alicante, Miguel Hernández-CSIC, San Juan de Alicante, 03550 Alicante, Spain; fnavarrete@goumh.umh.es (F.N.-R.); jmanzanares@umh.es (J.M.-R.); ma.cortes@umh.es (M.A.C.-G.); 4Centro de Investigación Biomédica en Red Sobre Enfermedades Neurodegenerativas (CIBERNED), San Juan de Alicante, 03550 Alicante, Spain; 5Institute of Sanitary and Biomedical Research of Alicante (ISABIAL), 03010 Alicante, Spain; sanchez_jos@gva.es; 6Preventive Medicine Service, University General Hospital of Alicante, 03010 Alicante, Spain; 7Neurology Department, University General Hospital of Alicante, 03010 Alicante, Spain; ruthgasparini@gmail.com (R.G.-B.); raquel.romero.lorenzo@gmail.com (R.R.-L.)

**Keywords:** Alzheimer’s, biomarker, Lewy body disease, mild cognitive impairment, α-synuclein

## Abstract

*Background:* Alzheimer’s disease (AD) is characterized by the presence of β-amyloid plaques and neurofibrillary tangles, while Lewy body dementia (LBD) is characterized by α-synuclein (α-syn) inclusions. Some authors examine α-syn protein in the neurodegeneration process of AD and propose to consider cerebrospinal fluid (CSF) α-syn as a possible additional biomarker to the so-called “core” of AD. *Objective:* To determine whether there is a correlation between α-syn levels and “core” AD biomarkers in patients with mild cognitive impairment (MCI). *Materials and methods:* In total, 81 patients in the early stages of MCI were selected from the outpatient dementia consultation in Alicante General Hospital. Using a cross-sectional case–control design, patients were analyzed in four groups: stable MCI (MCIs; *n* = 25), MCI due to AD (MCI-AD; *n* = 32), MCI due to LBD (MCI-LBD; *n* = 24) and a control group of patients with acute or chronic headache (Ctrl; *n* = 18). Correlation between CSF protein levels in the different groups was assessed by the Rho Spearman test. *Results:* We found positive correlations between T-tau protein and α-syn (*ρ* = 0.418; *p* value < 0.05) and p-tau_181p_ and α-syn (*ρ* = 0.571; *p* value < 0.05) exclusively in the MCI-AD group. *Conclusion:* The correlation found between α-syn and tau proteins in the first stages of AD support the involvement of α-syn in the pathogenesis of AD. This result may have clinical and diagnostic implications, as well as help to apply the new concept of “precision medicine” in patients with MCI.

## 1. Introduction

Alzheimer’s disease (AD) is the most frequent cause of dementia, followed by Lewy body dementia (LBD). Pathophysiologically, AD is typically characterized by the presence of β-amyloid plaques and neurofibrillary tangles, while LBD is characterized by α-synuclein (α-syn) inclusions. However, some authors examine α-syn protein in the neurodegeneration process of AD and propose to consider cerebrospinal fluid (CSF) α-syn as a possible additional biomarker to the so-called “core” of AD (Aβ42, T-tau and p-tau_181p_).

Quantification of α-syn protein in the CSF of AD patients is not usual practice. However, it has been studied by different authors [1,2,3,4,5,6,7,8,9]. Most of the published results indicate that CSF α-syn, in combination with the “core” AD CSF biomarkers, have clinical value in the differential diagnosis of AD and LBD at the stage of dementia, but few authors have not found a role for CSF α-syn as a useful biomarker for AD [3].

In experimental studies, the “in vivo” interaction between β-amyloid, tau and α-syn pathways has been demonstrated, promoting both aggregation and accumulation among themselves and accelerating cognitive dysfunction [10,11,12]. Moreover, there are some studies in humans that demonstrate this interaction in AD patients [2,13].

The aim of this study was to examine the relationship between α-syn levels with “core” AD CSF biomarkers in patients at the stage of mild cognitive impairment (MCI), who, after a long clinical follow-up, developed AD, LBD or remained stable.

## 2. Materials and Methods

### 2.1. Study Design and Participants

This cross-sectional monocenter case–control study was conducted in the outpatient dementia consultation of Neurology Service at the University General Hospital of Alicante (Spain) between 2008 and 2017. MCI was assessed following the Petersen criteria [14].

Patients’ inclusion criteria were: age over fifty-five, with concordant clinical and neuropsychological diagnosis, particularly an MMSE test score ≥ 24 and an IQCODE over 78. Informed consent was obtained before inclusion in this study and before performing the lumbar puncture (LP). Exclusion criteria were as follows: the presence of dementia or other neurological, psychiatric or medical disease, which could provoke cognitive deterioration; anticoagulant therapy; failure to obtain informed consent; and a score greater than five using Yesavage scale of depression.

All included patients underwent physical and neurological examination, neuropsychological study, cerebral magnetic resonance, blood test, LP and cerebral DAT-SCAN when LBD was suspected. Enrolled patients were reviewed every 6–12 months regarding the development of clinical dementia criteria during almost 4 years of follow-up.

The clinical criteria followed for considering conversion of MCI to AD (MCI-AD) were the NIA-AA criteria [15], while the McKeith criteria were used for conversion to LBD (MCI-LBD) [16]. Stable MCI (MCIs) was considered when patients remained free of dementia for almost 4 years after LP or until death.

Following these criteria, 24 patients were included in the MCI-LBD group with abnormal DAT-SCAN and non-AD CSF criteria. To avoid differences in age with the other MCI groups, 32 patients were selected in the MCI-AD group, fulfilling both the clinical criteria and the new biological NIA-AA criteria [17]. Likewise, 25 patients were included in the MCIs group. Finally, we included a control group (*n* = 18) composed of subjects with acute or chronic headache (*n* = 12) or pain syndromes (*n* = 6) who did not develop cognitive decline during a follow-up period of 4 years.

### 2.2. Obtention of CSF

All the CSF samples were obtained between 10:00 and 14:00. LP was performed by a neurologist with a 20 × 3.5 gauge needle. CSF was collected in standard tubes and centrifuged 10 min at 1500× *g* and then aliquoted in propylene tubes. Samples were stored at −80 °C until analyzed. Only those samples with fewer than 50 red blood cells/µL were included.

### 2.3. Measurement of Core AD CSF Biomarkers

Aβ1–42, T-tau and p-tau_181p_ were measured using a commercial ELISA kit (Innotest, Innogenetic/Fujirebio, Ghent, Belgium) following manufacturer´s instructions. Assays were tested blind with respect to clinical diagnosis.

### 2.4. Measurement of CSF α-Synuclein

Determination of α-synuclein levels in CSF was performed using the LEGEND MAX human α-synuclein ELISA kit with a precoated plate (BioLegend, San Diego, CA, USA), according to manufacturer´s instructions. This assay has been previously validated in a European-wide inter-laboratory study [18]. Luminiscence detection was carried out with BMG Labtech LUMIstar Optima. Assays were tested blind with respect to clinical diagnosis.

### 2.5. Statistical Analysis

The Kolmogorov–Smirnov test was used to analyze the distribution of each variable. A Student’s *t*-test for parametric variables and a Mann–Whitney U test for non-parametric variables were employed for comparisons between two groups. ANOVA was used for parametric variables and the Kruskal–Wallis test was used for non-parametric variables for comparisons between groups. Correlation between CSF protein levels was assessed by the Rho Spearman test. In all hypotheses, a *p* value of less than 0.05 determined statistical significance. The statistical package SPSS 21.0 was employed.

### 2.6. Ethical Criteria

The project was approved by the Ethical Committee of the University General Hospital of Alicante (ref. nº: PI2019/111).

## 3. Results

### 3.1. Clinical, Demographic Characteristics and Levels of CSF Biomarkers

The clinical and demographic features of each subject group were described in a previous publication [19]. In brief, no significant differences in age, gender, antecedents of hypertension, diabetes, hyperlipidemia, depression, Folstein MMSE score, onset of the symptoms, IQCODE score, years of schooling or NPI score were observed between the MCI groups. There was a small difference in age between the control group and the MCI groups and, in the clinical follow-up, that was larger in the stable MCI group than in the other groups. In the MCI-AD and MCI-LBD groups, Aβ42 levels were lower than in control and stable MCI patients, but Aβ42 did not discriminate between MCI-AD and MCI-LBD patients. T-tau and p-tau_181p_ levels were higher in MCI-AD patients compared to the other groups, with no differences between control, stable MCI and MCI-LBD. Finally, α-syn levels were statistical different between the four groups (Table 1).

### 3.2. Total Population

Taking the population as a whole, α-syn CSF levels were positively correlated with both T-tau (*ρ* = 0.509, *p*-value < 0.01) and p-tau_181p_ (*ρ* = 0.528, *p*-value < 0.01). As expected, there was a strong positive correlation between T-tau and p-tau_181p_ protein levels (*ρ* = 0.882, *p* value < 0.01) and a slight negative correlation between Aβ_1–42_ protein and T-tau (*ρ* = −0.475, *p*-value < 0.01) and p-tau_181p_ (*ρ* = −0.478, *p*-value < 0.01) (Figure 1).

### 3.3. MCI-AD Group

In this group, there were statistically significant correlations between α-syn levels with T-tau (*ρ* = 0.418, *p*-value < 0.01) (Figure 2B), particularly with p-tau_181p_ (*ρ* = 0.571, *p*-value < 0.001) (Figure 2C).

T-tau and p-tau_181p_ CSF levels were positively correlated (*ρ* = 0.511, *p*-value < 0.003) (Figure 2A) but no other correlations were found between the “core” biomarkers with either of the CSF α-syn levels.

### 3.4. Control Group

There was only a statistically significant correlation between T-tau and p-tau_181p_ in the control group (*ρ* = 0.797, *p*-value < 0.0001). The low correlation between α-syn and T-tau (*ρ* = 0.288) or p-tau_181p_ (*ρ* = 0.276) did not reach statistical significance in this group.

No other correlations were found between the “core” biomarkers with either of the α-syn CSF levels.

### 3.5. MCIs Group

In this group, there was a statistically positive correlation between T-tau and p-tau_181p_ (*ρ* = 0.562, *p*-value < 0.003). No other significant correlations were found between the “core” biomarkers in either of the α-syn CSF levels.

### 3.6. MCI-LBD Group

There was a strong positive correlation between T-tau and p-tau_181p_ (*ρ* = 0.822, *p* value < 0.0001) in this group. The correlations between α-syn and both T-tau (*ρ* = 0.21) and p-tau_181p_ (*ρ* = 0.29) were low and without statistical significance. No other significant correlations were found between the “core” biomarkers with either of the α-syn CSF levels.

## 4. Discussion

The most significant finding of this work was the strong relationship we found between T-tau and p-tau_181p_ with α-syn levels in the MCI-AD group. These results agree with previous publications in which a positive correlation between tau proteins and α-syn in AD patients was described [2,13]. However, these correlations disappeared in LBD patients and control subjects [2,13].

In animal models, α-syn has been related with the pathophysiology of AD due to its association with AD-triggering effectors Aβ_1–42_ and p-tau [10,11,12]. A synergic mechanism between them should lead to neurodegeneration [10,11,12].

Tau protein and α-syn share common properties and potentially even spreading mechanisms. Stabilization of microtubules, localization at the same cellular region, solubility and the shape of insoluble aggregates that provoke cellular dysfunction have been described; these similarities may explain their relationship in dysfunction and AD pathogenesis [20,21]. Otherwise, α-syn may initiate the fibrillization of tau and stimulate tangle production [22].

No relations were found between tau proteins and α-syn CSF levels in MCI-LBD patients in this study. However, some this relation has been previously described in a more advanced stage of the illness. Considering α-syn as a protein with a synaptic function, its higher levels could be related to more advanced neurodegeneration stages [4,23] in which different pathogenic pathways might be involved [2,4]. The synaptic loss may provoke an increase in tau protein and α-syn levels and subsequent clinical deterioration. However, the AD patients with a higher rate of progression showed an increase in p-tau and a decrease in α-syn CSF levels, suggesting a mixed pathophysiology [13]; these data may be important for prognostic purposes.

There is no agreement on whether the additional α-syn measurement to the “core” AD CSF biomarkers improves the differential diagnosis between AD and LBD [2,8,19]. In our previous publication [19], we found that diagnosis validity was high for both α-syn and p-tau_181p_, and their combination improved the discrimination of MCI-AD from the other three groups. For that reason, we and others [9] consider that combined measurements could provide a pathophysiological report that would be highly important for the new concept of “precision medicine” in degenerative dementia [24].

This is study has some limitations. First, this study is unicentric and we did not perform anatomopathological verification. Otherwise, DAT-SCAN was performed exclusively in MCI-LBD patients, but no clinical signs or symptoms of LBD were observed in the rest of the groups included during clinical follow-up.

## 5. Conclusions

The correlation found between α-synuclein and tau proteins in the first stages of AD corroborates the involvement of this protein in the pathogenesis of the illness. This result may have clinical and diagnostic implications, and it may be helpful in the application of “precision medicine” to patients with mild cognitive impairment.

## Figures and Tables

**Figure 1 medicina-57-00954-f001:**
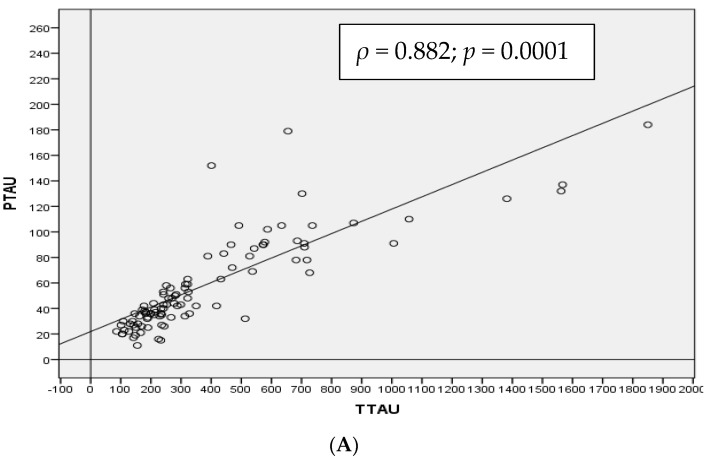
Spearman correlation (*ρ*) in the whole cohort between CSF T-tau and p-tau_181p_ (**A**), α-synuclein and T-tau (**B**) and α-synuclein and p-tau_181p_ (**C**) proteins in total population.

**Figure 2 medicina-57-00954-f002:**
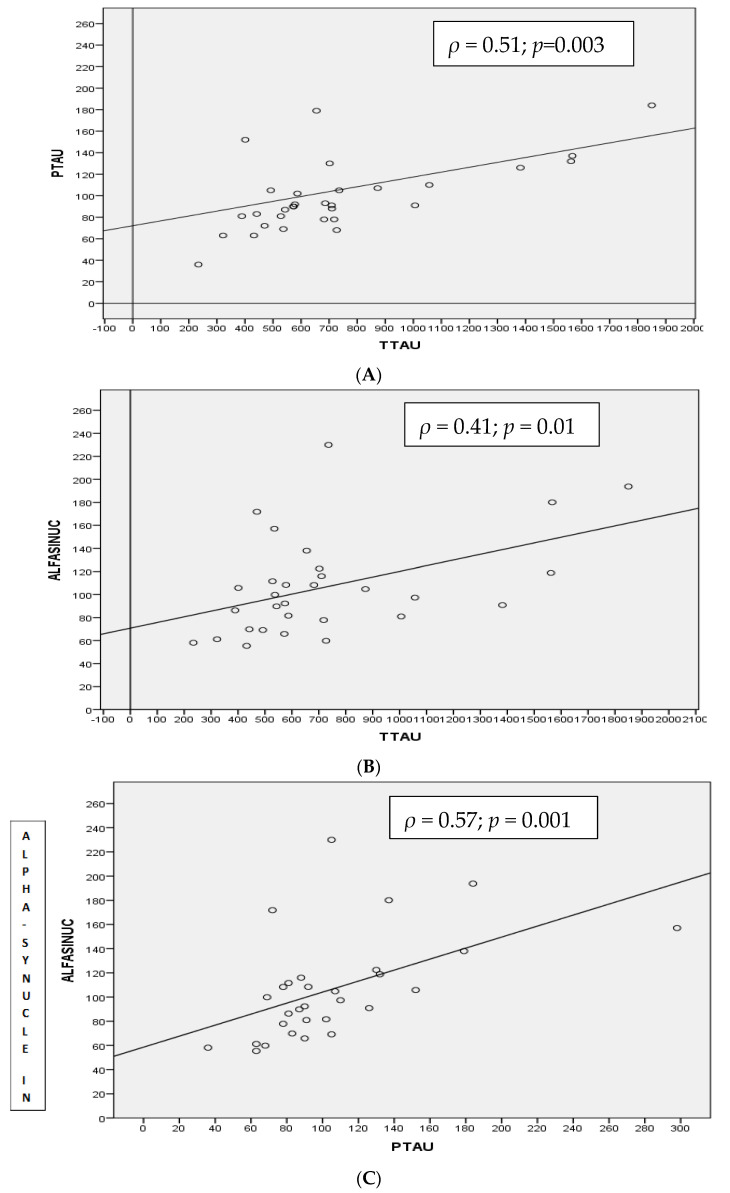
Spearman correlation (*ρ*) between CSF T-tau and p-tau_181p_ (**A**), α-synuclein and T-tau (**B**) and α-synuclein and p-tau_181p_ (**C**) proteins in MCI-AD group.

**Table 1 medicina-57-00954-t001:** Clinical, demographic characteristics and levels of CSF biomarkers in the AD, LBD, stable MCI and control groups.

	Control	Stable MCI	LBD	AD	*p* Value
**Number**	18	25	24	32	-----
**Age (mean ± SD)**	67.3 ±5.7	70.5 ± 7.3	74.7 ± 5.8	71.3 ± 7.3	0.007
**Males (%)**	8 (44)	12 (48)	12 (50)	12 (37.5)	0.7
**Antecedents (%):**					
**Diabetes**	7 (39)	5 (20)	3 (12)	6 (19)	0.06
**Hypertension**	11 (61)	14 (56)	15 (62)	15 (47)	0.8
**Hyperlipidemia**	14 (77)	18 (72)	11 (45)	14 (44)	0.07
**Depression**	8 (44)	18 (72)	13 (54)	15 (47)	0.06
**Schooling years (mean ± SD)**	4.5 ± 1.2	4.6 ± 1.2	3.8 ± 1.1	4.1 ± 0.9	0.7
**Start of the symptoms before LP months (mean ± SD)**	--------	19 ± 15	31 ± 18	18 ± 7	0.06
**MMSE (mean ± SD)**	27.5 ± 1.2	25.6 ± 1.6	24.5 ± 2.0	24.2 ± 2.4	0.5
**IQCODE (mean ± SD)**	--------	62.0 ± 8.6	68.7 ± 9.9	70.0 ± 7.6	0.08
**NPI (mean ± SD)**	-------	10.6 ± 2.5	9.1 ± 2.0	10.3 ± 1.9	0.8
**Clinical follow-up after LP months (mean ± SD)**	39 ± 21.6	80 ± 22	41 ± 33	41 ± 29	0.01
**Amnestic/non amnestic MCI**	-------	14/11	5/19	25/2	MCI/LBD: 0.5 MCI/AD: 0.01 AD/LBD: 0.01
**MRI-MTA** **(mean ± SD)**	1.1 ± 1.5	3.5 ± 1.1	3.7 ± 1.2	4.3 ± 1.5	0.01
**Aβ1–42 levels (pg/mL)** **median [p25-p-75]**	1274 [1139–1410]	925 [772–1078]	612 [436–743]	499 [430–583]	MCI/MCI-LBD: <0.001 MCI/MCI-AD: <0.001 MCI-AD/MCI-LBD: n.s.
**T-tau levels (pg/mL)** **median [p25-p-75]**	212 [171–236]	225 [157–282]	243 [159–292]	621 [510–732]	MCI/MCI-LBD: n.s. MCI/MCI-AD: <0.001 MCI-AD/MCI-LBD: <0.001
**p-tau_181p_ levels (pg/mL)** **median [p25-p-75]**	37 [34–40]	34 [27–42]	42 [25–52]	91 [80–118]	MCI/MCI-LBD: n.s. MCI/MCI-AD: <0.001 MCI-AD/MCI-LBD: <0.001
**α-syn levels (pg/mL)** **median [p25-p-75]**	1914 [1626–2664]	1505 [1219–1915]	936 [736–1121]	2557 [1985–3255]	MCI/MCI-LBD: <0.001 MCI/MCI-AD: <0.001 MCI-AD/MCI-LBD: <0.001

SD: standard deviation; MCI: mild cognitive impairment; LBD: Lewy body disease; AD: Alzheimer’s disease; MMSE: minimental state examination; IQCODE: informant questionnaire on cognitive decline in the elderly; NPI: neuropsychiatric inventory; MRI-MTA: magnetic resonance imaging-medial temporal atrophy. Significance levels are shown by the “*p* value” (n.s.: non-significant). (With permission from Wiley [19]).

## Data Availability

All the data are available from the corresponding author upon reasonable request.

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
