# Peer review of "Relation between Alpha-Synuclein and Core CSF Biomarkers of Alzheimer’s Disease"

_medicina, 2021, doi:10.3390/medicina57090954_

Round 1
Reviewer 1 Report
1. p-tau were measured using commercially ELISA kit (Innotest, In- 94nogenetic/Fujirebio, Ghent, Belgium)- Tau is a phosphorylated protein, containing 85 potential serine (S), threonine (T), and tyrosine (Y) phosphorylation sites. – Which phosphorylation sites is detected by ELISA kit used in this study- add this to the material and methods and in disscusion. In LBD the ptau is not the same as in AD so maybe that why you did not found the corelation
2. Is there a difference between the MCI of AD type and MCI of LBD in tau phosphorylation – add this to your results and discussion. Some literature data indicate that CSF phospho-tau (181P) may be a good marker for differentiation between AD and LBD and in your case MCI AD or LBD type.
Author Response
Thank you very much for your review and comments, that we response next:
- p-tau were measured using commercially ELISA kit (Innotest, In- 94nogenetic/Fujirebio, Ghent, Belgium)- Tau is a phosphorylated protein, containing 85 potential serine (S), threonine (T), and tyrosine (Y) phosphorylation sites. – Which phosphorylation sites is detected by ELISA kit used in this study- add this to the material and methods and in disscusion. In LBD the ptau is not the same as in AD so maybe that why you did not found the correlation. R: We measured p-tau 181p, that is in the kit from Fujirebio. We have added it in the material/methods and in dicussion. Effectively, the isoform p-tau may be different in LBD from AD.
2. Is there a difference between the MCI of AD type and MCI of LBD in tau phosphorylation – add this to your results and discussion. Some literature data indicate that CSF phospho-tau (181P) may be a good marker for differentiation between AD and LBD and in your case MCI AD or LBD type. R: We have added the CSF biomarker levels in the results and we have discussed the diagnosis validity of p-tau181p to differentiate between MCI-AD and MCI-LBD.
Yours
Dr. Monge Argilés.
Reviewer 2 Report
The authors were interested in the possible links between biomarkers classically measured in CSF in Alzheimer's disease and alpha-synuclein. The latter, which aggregates in synucleinopathies, is found increased in the CSF of AD patients. The authors show a positive correlation between alpha-synuclein and t-Tau or P-Tau in MCI AD patients.
The article is well written and easy to read. The discussion is adapted to the results obtained.
I would have just a few minor points :
- In figures 1 and 2, I guess ALFASINUC means alpha-synuclein, so please replace the f by ph and the i by a y.
- It would be interesting to add the mean values of the different biomarkers studied (Ab42, t-Tau, p-Tau and alpha-synuclein) in table 1 with the statistics.
- Please note also some typos in the references, please use a bibliography software to avoid this kind of problem.
Author Response
Thank you very much for your review and comments, that we response next:
- In figures 1 and 2, I guess ALFASINUC means alpha-synuclein, so please replace the f by ph and the i by a y. R: We agree with you. We have replaced it.
- It would be interesting to add the mean values of the different biomarkers studied (Ab42, t-Tau, p-Tau and alpha-synuclein) in table 1 with the statistics. R: We have included the CSF biomarker levels and the statistic in table 1.
- Please note also some typos in the references, please use a bibliography software to avoid this kind of problem. R: We have reviewed the references and corrected the mistakes.
- Yours
- Dr. Monge-Argilés.